# The Antioxidant and Anti-Inflammatory Properties of Bee Pollen from Acorn (*Quercus acutissima* Carr.) and Darae (*Actinidia arguta*)

**DOI:** 10.3390/antiox13080981

**Published:** 2024-08-13

**Authors:** Jeong-Eun Kwak, Joo-Yeon Lee, Ji-Yoon Baek, Sun Wook Kim, Mok-Ryeon Ahn

**Affiliations:** 1Department of Health Sciences, The Graduate School of Dong-A University, Busan 49315, Republic of Korea; kpj781120@naver.com (J.-E.K.); dlwndus523@naver.com (J.-Y.L.); byoon0403@gmail.com (J.-Y.B.); 2Research and Business Planning Team, Panolos Bioscience Inc., Hwaseong 18471, Republic of Korea; swkim@panolos.com; 3Center for Food & Bio Innovation, Dong-A University, Busan 49315, Republic of Korea

**Keywords:** bee pollen, antioxidant, anti-inflammatory, ROS, NF-κB

## Abstract

Aging is a complex biological process characterized by a progressive decline in physical function and an increased risk of age-related chronic diseases. Additionally, oxidative stress is known to cause severe tissue damage and inflammation. Pollens from acorn and darae are extensively produced in Korea. However, the underlying molecular mechanisms of these components under the conditions of inflammation and oxidative stress remain largely unknown. This study aimed to investigate the effect of bee pollen components on lipopolysaccharide (LPS)-induced RAW 264.7 mouse macrophages. This study demonstrates that acorn and darae significantly inhibit the LPS-induced production of inflammatory mediators, such as nitric oxide (NO) and prostaglandin E2 (PGE2), in RAW 264.7 cells. Specifically, bee pollen from acorn reduces NO production by 69.23 ± 0.04% and PGE2 production by 44.16 ± 0.08%, while bee pollen from darae decreases NO production by 78.21 ± 0.06% and PGE2 production by 66.23 ± 0.1%. Furthermore, bee pollen from acorn and darae reduced active oxygen species (ROS) production by 47.01 ± 0.5% and 60 ± 0.9%, respectively. It increased the nuclear potential of nuclear factor erythroid 2-related factor 2 (Nrf2) in LPS-stimulated RAW 264.7 cells. Moreover, treatment with acorn and darae abolished the nuclear potential of nuclear factor κB (NF-κB) and reduced the expression of extracellular signal-associated kinase (ERK) and c-Jun N-terminal kinase (JNK) phosphorylation in LPS-stimulated RAW 264.7 cells. Specifically, acorn decreased NF-κB nuclear potential by 90.01 ± 0.3%, ERK phosphorylation by 76.19 ± 1.1%, and JNK phosphorylation by 57.14 ± 1.2%. Similarly, darae reduced NF-κB nuclear potential by 92.21 ± 0.5%, ERK phosphorylation by 61.11 ± 0.8%, and JNK phosphorylation by 59.72 ± 1.12%. These results suggest that acorn and darae could be potential antioxidants and anti-inflammatory agents.

## 1. Introduction

Aging is a multifaceted biological process that entails a gradual reduction in physical function. With the increasing life expectancy and the advancement and enrichment of society, the prevalence of these inflammatory diseases is expected to rise. This process also leads to an increased vulnerability to age-related chronic diseases, such as cardiovascular diseases, cancer, and neurodegenerative disorders [1]. The phenomenon of age-associated chronic inflammation, termed ‘senoinflammation’, has garnered significant attention [2]. These chronic inflammations are closely linked to aging and age-related diseases. Aging induces changes at the molecular, cellular, and systemic levels in inflammation, resulting in senoinflammation [2].

Substances derived from natural sources, such as bee-pollen extract and phytochemicals, show great promise as anti-inflammatory treatments with reduced side effects of current therapeutics [3]. Currently, corticosteroids and non-steroidal anti-inflammatory medications (NSAIDs) are frequently used in conventional therapy regimens [4]. Nevertheless, these options have disadvantages, such as organ damage [4]. When long-term use of corticosteroid medications, such as prednisone, is suddenly discontinued, the body may not be able to produce hormones adequately, leading to steroid withdrawal symptoms, like stomach ulcers and elevated blood sugar. This drives the quest for natural substances that have fewer adverse effects, while simulating the effects of NSAIDs [5].

Bee pollen could be a promising alternative to current therapeutics like NSAIDs or corticosteroids by addressing current medical unmet needs. Recent studies confirm the potential of bee pollen or other bee products as antioxidants and anti-inflammatory medicines [6,7]. Bee pollen is a mixture of nectar and secretions from honeybees combined with floral pollen. It can be collected at the hive entrance, using traps [8]. Due to its anti-disease properties, bee pollen is incorporated into diets as an additional source of nutrients. It contains essential nutrients, including carbohydrates, proteins, fats, vitamins, and minerals, as well as trace elements and high concentrations of polyphenols—predominantly flavonoids [9]. The current research reported the antioxidant and anti-inflammatory effects of bee pollens [3,6,7], implying the potential of bee pollen for the development of antioxidant and anti-inflammation therapy without significant side effects [6,7]. It has played an important role in combating metabolic disorders such as diabetes, obesity, hyperlipidemia, and related cardiovascular complications, as well as anti-inflammatory antioxidants [10].

The most frequent cause of aging and chronic diseases associated with aging is oxidative stress. As aging progresses, reactive oxygen species (ROS), including hydrogen peroxide (H_2_O_2_) and hydroxyl radicals (^•^OH), accumulate and play a significant role in oxidative stress. They are known to induce DNA damage, lipid oxidation, protein denaturation, and destruction of cell membranes [6,11]. According to recent research, ROS can trigger inflammation by activating the mitogen-activated protein kinase (MAPK) signaling pathway, which triggers the release of a variety of inflammatory cytokines [12,13,14]. The expression of prostaglandin-endoperoxide synthase 2 (COX-2), nitric oxide synthase (iNOS), and nuclear factor κ-light-chain-enhancer of activated B cells (NF-κB) is triggered by ROS-induced c-Jun N-terminal kinases (JNKs) and IκB kinase (IKK) during hyperinflammation [15].

Inflammation is a complicated physiological reaction that shields the organism from externally damaging stimuli, including infections, particles, and viruses. It is a major mechanism mediating innate and adaptive immunity. Numerous chronic diseases, such as cancer, neurological, endocrine, metabolic, and cardiovascular conditions, are closely linked to the onset and progression of chronic inflammation due to the diverse and organ-specific cellular and molecular processes of chronic inflammation [16]. An inflammatory response occurs locally when a damaged tissue triggers the production of inflammatory mediators, such as cytokines like COX-2, (Cyclooxygenase-2), iNOS, and NF-κB, by means of many immune system components [17]. TNF-α (tumor necrosis factor alpha), which is mostly expressed in activated macrophages, is induced by immunogenic chemicals such as lipopolysaccharides (LPSs) and becomes elevated. Major oxidation and inflammatory agents, including pro-inflammatory cytokines, COX-2, iNOS, and nitric oxide (NO), are activated by NF-κB [18]. If inflammation persists due to these factors, as previously mentioned, normal tissue can be damaged and develop into chronic inflammation, leading to organ dysfunction syndrome.

Numerous studies have investigated bee pollen, yet the variety of the origin plant types and their regional specificities still remain underexplored. It is the first scientific report of these effects of bee pollens, specifically from acorn and darae. To our knowledge, no other research has focused specifically on bee pollen from acorn and darae. Previous reports have demonstrated the antioxidant and anti-inflammatory effects of bee pollen worldwide [19,20,21,22]. However, these results indicate that the effects of bee pollen vary slightly. This variation may be attributed to the nutritional and chemical properties of bee pollen, which are influenced by the origin plants and harvest regions. Previously, we analyzed the chemical properties of Korean bee pollen and found that they differ depending on the origin plants and harvest regions [23]. Based on our ongoing nutritional analysis (ongoing preparation for publication), the components and nutritional content of bee pollen vary with plant species and harvest regions. We speculate that the nutritional and chemical components and nutrition of bee pollen account for their various effects and characteristics of bee pollen.

In this study, we expand the research field of bee pollen from acorn and darae by exploring their anti-inflammatory and antioxidant properties. Current research indicates that pollens from acorn and darae exhibit negligible anti-inflammatory benefits [24,25] and no antioxidant properties. Using RAW 264.7 macrophages activated with LPS, we show the anti-inflammatory and antioxidant mechanisms of ethanol extracts of bee pollen (EEBP) from acorn and darae, which might have broad implications for the development of antioxidants and anti-inflammatory agents for aging-related diseases.

## 2. Materials and Methods

### 2.1. Materials

Fetal bovine serum (FBS), Dulbecco’s Modified Eagle’s medium (DMEM), and phosphate-buffered saline 1x (PBS 1x) were obtained from HyClone (HyClone, South Logan, UT, USA). Penicillin streptomycin, dimethyl sulfoxide (DMSO), lipopolysaccharides from *Escherichia coli* O111:B4 (LPS), Griess reagent, sodium nitrite, DAPI (4′,6-diamidino-2-phenylindole) readymade solution, 2′,7′-Dichlorofluorescin diacetate, and triton X-100 were purchased from Sigma (Sigma, St. Louis, MO, USA). Polyvinylidene difluoride (PVDF) blotting membranes and Western-blotting detecting solution of enhanced chemiluminescence (ECL) were obtained from GE Healthcare (GE Healthcare, Chalfont St. Giles, UK). A mounting medium with DAPI-aqueous fluoroshield was brought from Abcam (Cambridge, UK), and 4% paraformaldehyde was obtained from T&I (T&I, Chuncheon, Republic of Korea). Phospho- or total forms of IĸBα, NF-ĸB p-65, PI3K, Akt, ERK 1/2, JNK, and β-actin were purchased from Cell Signaling Technology (Danvers, MA, USA). Alexa fluor™ 488 goat anti-rabbit IgG from Invitrogen (Invitrogen, Waltham, MA, USA) and all other chemicals not mentioned were purchased from Sigma-Aldrich (Sigma-Aldrich, St. Louis, MO, USA).

### 2.2. Preparation of Bee Pollen

The acorn (*Quercus acutissima* Carr.) and darae (*Actinidia arguta*) pollen samples were collected in South Korea. The crude bee-pollen materials were kept frozen at −20 °C, and after rapid freezing at −80 °C in a deep freezer (DF8510, Ilshin Bio Base Co., Ltd., Gwangju, Republic of Korea), the freeze dryer (FD-5N, Tokyo Rikakikai Co., Ltd., Tokyo, Japan) was dried at the same temperature. The component of the pollens was extracted with 1.5 L of 70% ethanol at 200 g of each sample for 72 h, and the extracts were filtered out of filter paper. The ethanol suspensions were separated by centrifugation at 12,000 rpm for 5 min under 4 °C, and the supernatants were concentrated under reduced pressure to give ethanol extracts of bee pollen (EEBP). Then, the filtrates were concentrated under reduced pressure on an evaporator (N-100, Eyela, Tokyo, Japan), frozen, dried again, and converted into powder form. The EEBP was stored under dry conditions at 4 °C, until analysis.

### 2.3. Cell Culture and LPS Stimulation

RAW 264.7 cells as a mouse macrophage cell line were purchased from American Type Cell Culture (ATCC, Rockville, MD, USA) and then cultured in Dulbecco’s Modified Eagle Medium supplemented with 10% fetal bovine serum and 1% penicillin–streptomycin. Cells were maintained in a cultured, humidified atmosphere, with 5% CO_2_, at 37.5 °C. In all experiments, cells were used after stabilization and were incubated with samples pretreated 1 h before LPS treatment of 1 μg/mL

### 2.4. Cell-Viability Assay

For the evaluation of the cytotoxic effect of the acorn and darae pollen against RAW 264.7 cells, cell viability was measured using the Cell Counting Kit-8 (Dojindo, Kumamoto, Japan). Cell Counting Kit-8 (CCK-8) allows sensitive colorimetric assays for the determination of cell viability in cytotoxicity assays. RAW 264.7 cells (5 × 10^4^ cells/well) were seeded in a 96-well plate and incubated overnight. They were then pretreated with various concentrations of pollen samples (50–400 μg/mL) for 1 h and co-treated with 1 μg/mL LPS and pollen samples for another 24 h. The positive control was cells that were not treated, and the negative control was LPS. Then, 10 μL of a solution reagent containing WST-8 [2-(2-methoxy-4-nitrophenyl)-3-(4-nitrophenyl)-5-(2,4-disulfophenyl)-2H-tetrazolium, monosodium salt] was added to the wells of a 96-well plate containing samples in each 100 μL of culture medium, the plate was incubated at 37 °C for 4 h. The absorbance at 450 nm was recorded by a microplate reader (Bio-Rad Laboratories, Hercules, CA, USA).

### 2.5. Intracellular Reactive Oxygen Species (ROS) Assay

Intercellular reactive oxygen species (ROS) formation was detected by 2′,7′-Dichlorofluorescin diacetate (DCF-DA), which is oxidized to fluorescent 2′,7′-Dichlorofluorescin (DCF) by oxidative stress. RAW 264.7 cells were seeded at 1 × 10^4^ cells/well in a 24-well plate, incubated for 24 h, and further incubated during 72 h with various concentrations (50, 100, 200, and 400 μg/mL) of pollen samples. The positive control was cells that were not treated, and the negative control was LPS. We checked the ROS production time-dependently (24, 48, and 72 h). Based on that, we chose one time point, 72 h. After incubation for 30 min and washing with PBS 2 times, the cells were fixed with 4% paraformaldehyde (PFA) at 4 °C permeabilizated using PBS containing 0.1% triton solution. After being washed twice with PBS each step, the plates were incubated with 4′,6-diamidino-2-phenylindole (DAPI) solution for 15 min in a dark room at room temperature. Each well was washed twice with PBS and then incubated with DCF-DA for 30 min in a dark room at room temperature. The cells were observed in a Leica DMi8 fluorescent microscope (Leica, Wetzlar, Germany) under excitation and emission wavelengths of 485/535 nm at 200× magnification.

### 2.6. Immunofluorescence (IF) Assay

For a 24-h period, RAW 264.7 cells were cultured on glass coverslips to examine the nuclear localization of NF-κB and Nrf2 in 24-well plates. The positive control was cells that were not treated, and the negative control was LPS. Following a 1 h pretreatment with the sample, these cells were exposed to 1 µg/mL LPS for 3 h. After treatment, the cells were fixed in 4% paraformaldehyde for 15 min, permeabilized with 0.5% Triton X-100 in phosphate-buffered saline (PBS) for 15 min, and then blocked for 1 h at room temperature in PBS containing 5% BSA. After that, cells were treated with anti-NF-κB and Nrf2 antibodies (1:400 in 2.5% BSA in PBS-T) and left overnight at 4 °C. The cells were then treated at room temperature with the secondary antibody (goat anti-rabbit IgG cross-absorbed secondary antibody conjugated to AF 488). Following a PBS-T rinse, cells were counterstained for 20 min at room temperature using DAPI solution (2.5 µg/mL). Using a Carl Zeiss LSM700 laser scanning confocal microscope (Carl Zeiss, Oberkochen, Germany), fluorescent pictures were captured following PBS washing and slide mounting.

### 2.7. Cell-Morphology Assay

A cell-morphology assay was performed to observe the effect of the Acorn (*Quer cus acutissima* Carr.) and Darae (*Actinidia arguta*) pollen samples on the LPS-induced RAW 264.7 cells. The cells were seeded at 1 × 10^4^ cells/well into 24-well plates and were pretreated with pollen samples for 1 h and LPS for an additional 24 h. The positive control was cells that were not treated, and the negative control was LPS. For morphological evaluation, cells were additionally cultured with DMEM medium (control) and DMEM medium containing LPS (positive control), and Leica DMi8 phase-contrast microscopy (Leica, Wetzlar, Germany) was used to obtain images of the morphology at 18 h of each treatment under magnification (×200).

### 2.8. Nitric Oxide (NO) Production Assay

For the evaluation of NO production in the culture supernatant, nitrite, an oxidative product of NO, was measured. RAW 264.7 cells at 1 × 10^5^ cells/well were cultured in 96-well plates for 24 h and then incubated with LPS with or without pretreatment with pollen samples for 1 h. The positive control was cells that were not treated, and the negative control was LPS. After 18 h of culture, the culture supernatant was collected. The 100 μL culture supernatant was mixed with an equal volume of Griess reagent and incubated for 15 min at room temperature. Finally, the absorbance was measured at 540 nm, and the production of nitrite was calculated by comparison with the standard curve of sodium nitrite dissolved in DMEM.

### 2.9. Prostaglandin E2 (PGE2) Production Assay

PGE2 production was measured in culture medium in order to determine COX-2 activity. For the assay of COX-2 induction, RAW 264.7 cells (1 × 10^6^ cells/well) were seeded onto a 48-well plate and incubated overnight; they were then pretreated with samples for 1 h and co-treated with 1 µg/mL of LPS and pollen samples for another 24 h. The positive control was cells that were not treated, and the negative control was LPS. Centrifugation at 3000× *g* for 5 min under 4 °C was used to collect the culture supernatant. The amount of the Prostaglandin E2 (PGE2) production was measured by the Prostaglandin E2 Parameter Assay Kit (R&D Systems, Minneapolis, MN, USA). The experiments were conducted according to the manufacturer’s instructions, and absorbance at 450 nm was recorded by a microplate reader.

### 2.10. Enzyme-Linked Immunosorbent (ELISA) Assay

Pro-inflammatory cytokines, including TNF-α, IL-1β, and IL-6, were measured using an ELISA assay. Briefly, RAW 264.7 cells were plated onto a 48-well plate (1 × 10^6^ cells/well) overnight; the cells were pretreated with pollen samples (50, 100, 200, and 400 μg/mL) for 1 h and then treated with 1 µg/mL of LPS for 18 h. The positive control was cells that were not treated, and the negative control was LPS. The culture supernatant was harvested by centrifugation at 3000× *g* for 5 min under 4 °C. TNF-α, IL-1β, and IL-6 concentration were measured by the Mouse Quantikine ELISA Kit (R&D Systems, Minneapolis, MN, USA) according to the manufacturer’s instructions. Absorbance at 540 nm was recorded by a microplate reader.

### 2.11. Western Blot Analysis

RAW 264.7 cells (6 × 10^5^ cells/well) were seeded in a 6-well plate for 24 h. Various concentrations of pollen samples were pretreated for 1 h and LPS-stimulated (1 µg/mL) for an additional 24 h. The positive control was cells that were not treated, and the negative control was LPS. The cell pellets were harvested, and cells were lysed with ice-cold lysis buffer and centrifuged at 13,000 rpm for 20 min. The protein concentration of each sample was quantified to 20 μL of each sample by using the BCA protein assay kit (Thermo Fisher Scientific, Waltham, MA, USA), and the protein was boiled at 95 °C for 10 min. A total of 20 µL of each sample was loaded onto 9% sodium dodecyl sulfate–polyacrylamide gel electrophoresis (SDS-PAGE) and electrophoresed and then transferred to poly vinylidene membranes (PVDF). Immunoreactive protein bands were reacted using an enhanced chemiluminescent (ECL) Western-blotting detecting solution, according to the manufacturer’s directions. Band intensity was quantified and graphed via the ImageJ program (NIH, Bethesda, MD, USA). Results were obtained from three independent experiments.

### 2.12. Cell Cycle Assay

Cell cycle was measured using a Muse™ Cell Cycle kit (Merck Millipore, Darmstadt, Germany) according to the manufacturer’s instructions. Briefly, RAW264.7 cells were seeded on a 6-well plate (1 × 10^6^ cells/well), pretreated with pollen samples for 1 h, and then treated with 1 µg/mL of LPS for 24 h. The positive control was cells that were not treated, and the negative control was LPS. The cells were harvested by pipetting and washed twice with PBS. For cell fixation, samples were centrifuged at 3000× *g* for 5 minutes. After removing the supernatant, the cells were fixed with 70% ethanol at −20 °C for 3 h. The fixed cells were washed with PBS, added to 200 µL Cell Cycle Reagent, and then incubated at 37 °C in a CO_2_ incubator for 30 min in the dark. The cell cycle was analyzed using the Muse™ Cell Analyzer with the Muse 1.3.1 analysis program (Merck Millipore, Germany).

### 2.13. Statistical Analysis

All data were analyzed by SPSS version 27.0 software (Chicago, IL, USA). Data were presented as the mean ± standard deviation (SD) of three independent experiments. Throughout this study, we performed at least three independent experiments (for biological repetition) with duplicated samples (for technical repetition). One-way analysis of variance (ANOVA), followed by the Holm–Sidak method, was used to compare the data in different groups. Student’s *t*-test was conducted to compare control and test groups. Statistical significance was expressed with * *p* < 0.05, ** *p* < 0.01, and *** *p* < 0.001.

## 3. Results

### 3.1. Effect of EEBP from Acorn and Darae on Cell Viability and Cytotoxicity in RAW 264.7 Macrophages

To investigate whether EEBP from acorn and darae influence the viability and metabolism of the cells, we measured cell viability and morphology after RAW 264.7 cells were cultured for 24 h and treated with 0, 50, 100, 200, and 400 μg/mL of EEBP from acorn and darae, respectively. As a result, even at doses of up to 400 μg/mL of EEBP from acorn and darae, no cytotoxic effects were observed in these cells during the experimental period (Figure 1A,B). Thus, we employed up to 400 μg/mL of EEBP from acorn and darae for our next trials. RAW 264.7 cells were grown in media alone or treated with LPS for 24 h with or without EEBP from acorn and darae. Cell morphology was assessed under a microscope to investigate the effects of EEBP from acorn and darae on macrophage morphology (Figure 1C). As shown in the picture, normal cells were soft and round, while cells treated with LPS-stimulated RAW 264.7 cells changed into irregular, rough shapes with dendritic formation. Cells treated with EEBP from acorn and darae reduced the level of dendritic formation. Cells are visualized under an optical microscope (Figure 1C). As expected, Figure 1 showed that EEBP from both acorn and darae has no negative effects on the cell viability of RAW 264.7 cells.

### 3.2. Antioxidant Effect of EEBP from Acorn and Darae on ROS Production in RAW 264.7 Macrophages

RAW 264.7 cells were co-incubated with LPS and EEBP from acorn and darae for 24 h to evaluate the quantity of ROS generated by LPS exposure and to assess the capacity of EEBP from acorn and darae to mitigate these effects. By using DCFH-DA, ROS generation was quantified and identified by flow cytometry. Whereas LPS alone significantly increased the amount of ROS (Figure 2A), the treatment of EEBP from acorn and darae considerably reduced ROS production in LPS-pretreated RAW 264.7 cells in a dose-dependent manner (Figure 2A,B). First, at concentrations of 50, 100, 200, and 400 μg/mL, acorn reduced the intracellular ROS levels to 424. 2%, 338.1%, 317.5%, and 236.3%, respectively. And at concentrations of 50, 100, 200, and 400 μg/mL, darae reduced the intracellular ROS levels to 365. 0%, 282.0%, 222.5%, and 175.0%, respectively. These data demonstrated that LPS-induced ROS generation was inhibited by acorn and darae, implying the therapeutic effects of EEBP from acorn and darae as an antioxidant (Figure 2A,B).

### 3.3. Effect of EEBP from Acorn and Darae on the Nrf2 Pathway in RAW 264.7 Macrophages

When exposed to oxidative stimulation, the transcription factor Nrf2 functions as a sensor for oxidative stress and regulates the gene expressions involved in the antioxidant stress response. We hypothesized that the activation of antioxidant genes through Nrf2 mediated the protective effects of EEBP from acorn and darae against LPS-induced oxidative stress (intracellular ROS buildup). Using an immunofluorescence assay, we tracked the nuclear translocation of Nrf2. As depicted in Figure 3, when compared to the control cells, EEBP from acorn and darae dramatically enhanced the nuclear accumulation of Nrf2. Specifically, after the administration of EEBP from acorn and darae, Nrf2 accumulation was observed in the nucleus, whereas in LPS-treated cells, Nrf2 expression was predominantly found in the cytoplasm (Figure 3A,B). These findings strongly suggest that EEBP from acorn and darae promotes Nrf2 activation and nuclear translocation in macrophages.

### 3.4. Effect of EEBP from Acorn and Darae on Pro-Inflammatory Mediators in RAW 264.7 Macrophages

The effects of EEBP from acorn and darae on LPS-induced NO and PGE2 generation in RAW 264.7 macrophages were investigated by using Griess reagent and ELISA analyses, respectively. For 24 h, the cells were pretreated with LPS and different concentrations of EEBP from acorn and darae. In Figure 4A,B, the LPS-stimulated RAW 264.7 cells upregulated the expression of NO and PGE2 in comparison to the non-LPS treatment group. The LPS-induced overexpression of NO and PGE2 in RAW264.7 cells was significantly inhibited by pretreatment with EEBP from acorn and darae in a dose-dependent manner.

To examine the enzyme activities of iNOS and COX-2 that mediate the synthesis of NO and PGE2, we performed a Western blot analysis to measure the expression of iNOS and COX-2. As demonstrated in Figure 4C, iNOS and COX-2 protein levels were undetectable in the absence of LPS stimulation. The treatment with LPS alone markedly increased the expression of iNOS and COX-2. However, the pretreatment with EEBP from acorn and darae dramatically dose-dependently reduced the LPS-stimulated overexpression of iNOS and COX-2 proteins in RAW264.7 cells. According to these findings in Figure 4, EEBP from acorn and darae was able to suppress the expression of the iNOS and COX-2 enzymes, resulting in a consequent decrease in the generation of NO and PGE2.

### 3.5. Effect of EEBP from Acorn and Darae on Pro-Inflammatory Cytokines in RAW 264.7 Macrophages

To examine how EEBP from acorn and darae affect the generation of pro-inflammatory cytokines, we performed ELISA assays to assess the amounts of TNF-α, IL-1β, and IL-6 in the culture media from RAW264.7 cells treated with LPS with or without bee pollens from acorn and darae. As a result, the data shown in Figure 5 illustrate that the pretreatment with EEBP from acorn and darae significantly decreased the release of TNF-α, IL-1β, and IL-6 in a concentration-dependent manner, whereas the challenge with LPS alone caused increases in the levels of these three cytokines. However, there is no significant difference between the EEBPs from acorn and darae in pro-inflammatory cytokines.

### 3.6. Effect of EEBP from Acorn and Darae on NF-κB in RAW 264.7 Macrophages

The transcription of pro-inflammatory genes is regulated by NF-κB activation through post-translational modification and nucleus translocation in response to inflammatory stimulation. To ascertain how bee pollen influences the NF-κB p65 subunit’s translocation, RAW 264.7 cells were treated with EEBP from acorn and darae, and then they were stimulated with LPS. Subsequently, Western blot analyses were performed to observe the activities of NF-κB p65 and IκB-α. Figure 6A–C demonstrates that LPS-stimulated cells showed higher expression levels of the proteins p65 and IκB-α. However, LPS-stimulated nuclear accumulation of p65 and IκB-α was significantly diminished by treatment with EEBP from acorn and darae in a concentration-dependent manner.

Immunocytochemistry analysis also revealed that NF-κB p65 was typically sequestered in the cytoplasm under LPS treatment, which is consistent with the Western blot analysis (Figure 6A–C). However, pretreatment with EEBP from acorn and darae significantly inhibited LPS-mediated nuclear translocation of NF-κB (Figure 6D,E).

### 3.7. Effect of EEBP from Acorn and Darae on MAPKs in RAW 264.7 Macrophages

Using phospho-specific antibodies against the MAPK protein, we examined the effect of EEBP from acorn and darae on the MAPK pathways. The phosphorylation of three of the APK proteins, ERK, JNK, and p38, was upregulated by LPS stimulation (Figure 7). Furthermore, EEBP from acorn and darae reduced the LPS-induced phosphorylation of ERK, JNK, and p38 MAPK in a concentration-dependent manner.

### 3.8. Effect of EEBP from Acorn and Darae on Cell Cycle in RAW 264.7 Macrophages

We also investigated whether cell cycle arrest could be involved in the suppressive effect of bee pollen. Figure 8 demonstrates that the S phase decreased from 22.2% to 19.4%, the G1 phase increased from 53.5% to 63.9%, and the G2/M phase decreased from 24.26% to 16.8% in both groups’ LPS treatment groups with LPS-induced inflammation. These findings suggested that G1 arrest could result from an inflammatory response triggered by LPS. However, cells pretreated with bee pollens from acorn and darae exhibited a dose-dependent decrease in the proportion of G1 cycles. The S phase showed minimal change in the bee pollen-treatment group compared to the LPS-treated group.

## 4. Discussion

In this study, we investigated the antioxidant and anti-inflammatory effect of bee-pollen components from acorn and darae, which are predominantly produced and consumed in Korea. By using LPS-induced RAW 264.7 mouse macrophages, we demonstrated the inhibitory effects of EEBP from acorn and darae on the LPS-induced production of inflammatory mediators, such as NO and PGE2. EEBP from acorn and darae reduced ROS production, decreased the activities of NF-κB, and diminished the expression of ERK and JNK phosphorylation, whereas they increased Nrf2 in LPS-stimulated RAW 264.7 cells.

The effects of EEBP from acorn and darae were investigated using the murine macrophage cell line RAW 264.7 to evaluate their immunomodulatory properties. According to a recent study, macrophages perform a variety of tasks and functions. By coordinating a range of biological activities, they play an important role in maintaining homeostasis and normal physiological conditions [26]. In vitro assay using cell lines as a preliminary screening tool for biological activity is essential in studies investigating the potential immunological properties of natural compounds and their derivatives [27]. For our current study, we selected RAW 264.7 macrophages due to their effectiveness as targeted single cells for evaluating immunological reactivity [28,29,30,31]. LPS can trigger the production of pro-inflammatory chemicals, including NO and PGE2, by activating intracellular signaling pathways, such as the MAPK and NF-κB signaling pathways in macrophages [32]. Many extracellular stimuli can activate pathways that induce NF-κB, leading to phosphorylation and subsequent proteasomal degradation of the inhibitory molecule, IκB [33]. This process allows NF-*κ*B to translocate to the nucleus [34], where it binds to specific DNA binding sites to regulate transcription of over 150 genes, including stress-response proteins, proinflammatory cytokines, antimicrobial peptides, chemokines, and antiapoptotic proteins [35]. Our Western blot analysis revealed that both p65 and pIκB-α were significantly reduced, and immunofluorescence staining revealed that the translocation from cytoplasm to nucleus was suppressed upon treatment with pollens from acorn and darae.

In this study, we demonstrated that bee pollen from acorn and darae is involved in negatively regulating multiple inflammatory responses. One of the key enzymes catalyzing the conversion of L-arginine into nitric oxide (NO) is inducible nitric oxide synthase (iNOS) [36]. iNOS plays a significant role in eliminating bacterial illnesses and parasites, with the regulation of NO production primarily occurring at the transcriptional level. It has been demonstrated that LPS, cytokines, and oxidative stress activate NF-κB, which subsequently causes iNOS expression [37,38]. Cyclooxygenase-2 (COX-2) is another critical component of the inflammatory response, and its expression is rapidly induced by various external and intracellular stimuli, including LPS [39]. NF-κB has been shown to regulate the induction of COX-2 transcription [40,41]. Our finding indicates that the transcription of iNOS and COX2 was significantly reduced by pollens from acorn and darae. These pollens from acorn and darae also reduced NO generation in a dose-dependent manner in LPS-stimulated RAW 264.7 cells. We conclude that pollens from acorn and darae suppress LPS-induced inflammatory gene expression by reducing NF-κB transcriptional activity, as NF-κB is essential for controlling the production of iNOS and COX-2. Furthermore, tumor necrosis factor alpha (TNF-α) is a significant cytokine associated with the inflammatory response, playing an essential role in many acute and chronic inflammatory conditions. Interleukin-1beta (IL-1β) and interleukin-6 (IL-6) are also known as vital mediators in inflammation [42,43,44]. Thus, this study proves the anti-inflammatory effects of bee pollens from acorn and darae by demonstrating that bee pollens from acorn and darae significantly suppress the LPS-induced overexpression of NO, PGE2, TNF-α, IL-1β, and IL-6 in a dose-dependent manner.

Our findings reveal that bee pollens from acorn and darae reduces ROS levels. ROS, functioning as endogenous signaling molecules, regulate multiple signaling pathways involved in various intracellular processes that are critical for the activation of Nrf2 [45]. To further understand the potential of bee pollen from acorn and darae, we investigated the molecular mechanism of action underlying their anti-inflammatory activity by examining the effects of bee pollen from acorn and darae on ROS release and MAPK phosphorylation.

Our results showed that EEBP from acorn and darae promotes ROS release in RAW264.7 cells, which, in turn, activates the nuclear translocation of Nrf2, exerting an anti-inflammatory effect. The activation of MAPK is known to induce Nrf2-mediated HO-1 expression [46]. Moreover, ROS has been reported to mediate the activation of various kinases, including MAPK [1]. The treatment of bee pollen from acorn and darae significantly inhibited LPS-induced phosphorylation of ERK, JNK, and p38 proteins in RAW264.7 cells. These results indicate that the effect of bee pollen from acorn and darae is mediated through Nrf2 activation via the ROS-dependent MAPKs pathway. In a recent paper, when treated with ethanol-extracted acorns (EeA), they inhibited the expression of interleukin (IL)-8 stimulated by induction nitric oxide synthase (iNOS), cyclooxygenase 2 (COX2), monocyte chemoattractant protein-1 (MCP-1), and tumor necrosis factor alpha (TNF-α) in human-adult low-calcium (HaCaT) cells [27]. The difference from our paper is the additional investigation of the monocyte chemotrained protein-1 (MCP-1) pathway, a type of chemokine that moves and attaches monocytes, which play an important role in the early stages of atherosclerosis, as well as anti-inflammation.

Despite the extensive research on bee pollen, the diverse origin plant types of pollen and regional characteristics have not been thoroughly examined. This study investigated the characteristics of bee pollen from acorn and darae in South Korea in antioxidation and anti-inflammation. We previously researched the chemical properties of Korean bee pollen and their catechol-O-methyltransferase inhibitory activities, indicating their potential prevention and treatment of Parkinson’s disease and depression [42]. This study broadens our understanding by demonstrating the effects of bee pollen from acorn and darae, supporting their development as an effective anti-inflammatory and antioxidant treatment. Enhancing the potential development of bee pollen can stimulate its consumption and use in functional foods and medicine. Consequently, this research will benefit the apiculture industry and the local economy. Therefore, this study will contribute to both scientific knowledge and the local and apiculture industry.

## 5. Conclusions

This study provides a comprehensive understanding of the molecular mechanisms underlying the antioxidative and anti-inflammatory actions of bee pollens from acorn and darae in RAW 264.7 macrophages. Bee pollens from acorn and darae significantly inhibit pro-inflammatory cytokines, such as TNF-α, IL-6, and IL-1β, and the NF-κB and MAPK signaling pathways. Furthermore, this study revealed that bee pollen reduced inflammation by blocking NF-κB nuclear translocation and triggering Nrf2 signaling pathways that are dependent on ROS and MAPKs. However, there is no significant difference in the antioxidant and anti-inflammatory effects of EEBP from acorn and darae. All in all, these results shed light on the development of pollens from acorn and darae as antioxidants and anti-inflammatory agents.

## Figures and Tables

**Figure 1 antioxidants-13-00981-f001:**
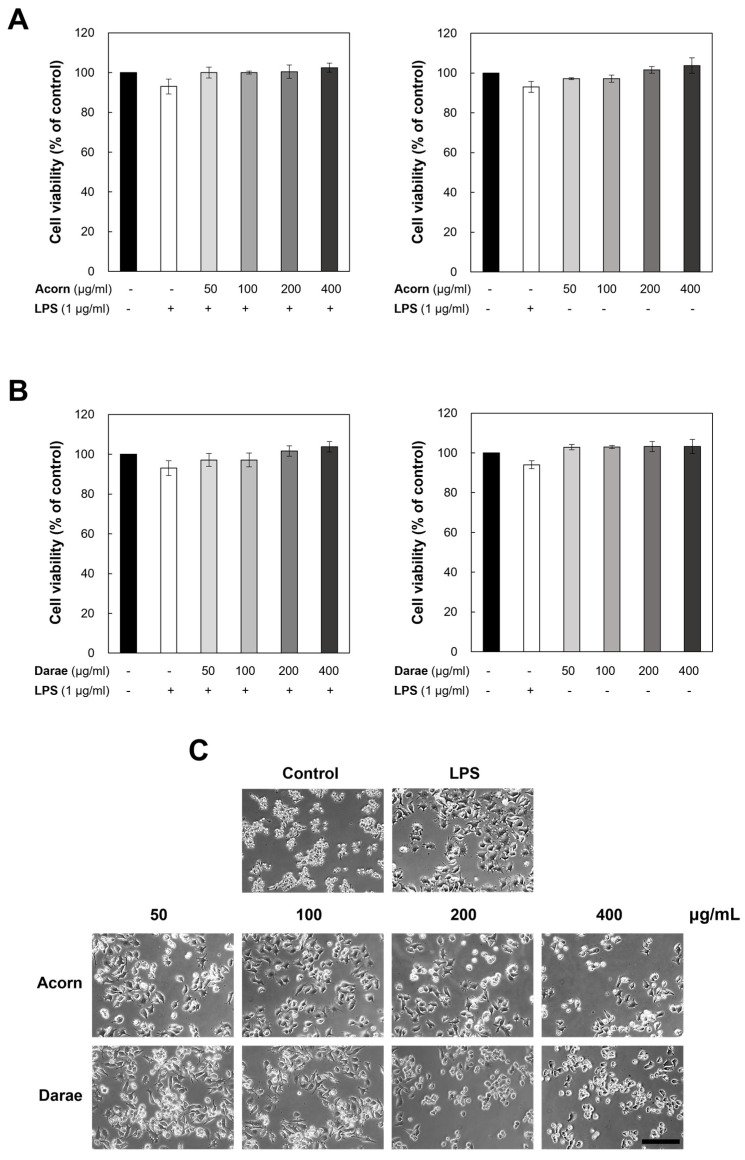
EEBP from acorn and darae does not affect the cell viability and growth of RAW 264.7 cells. (**A**,**B**) The MTT test was used to assess the viability of the cells after they were exposed to different doses of EEBP from acorn and darae for 24 h. The data were obtained from three separate experiments and are shown as the mean ± SD. (**C**) Raw 264.7 cell morphological alterations were observed under treatment of EEBP from acorn and darae.

**Figure 2 antioxidants-13-00981-f002:**
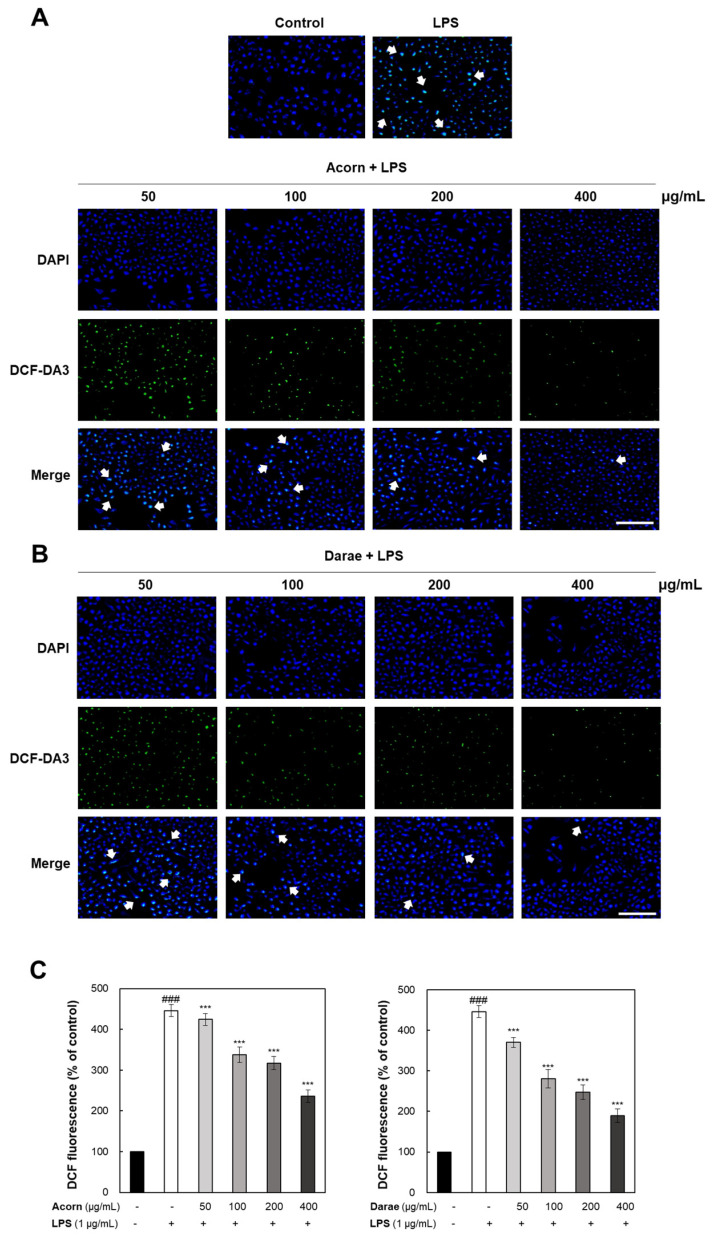
EEBP of acorn and darae inhibits LPS-induced ROS generation in RAW 264.7 cells. LPS (1 µg/mL)-stimulated RAW 264.7 cells were treated with various concentrations of EEBP from acorn and darae and incubated for 24 h. DCFH-DA was then applied for 30 min. The white arrow points to representative examples of the co-localization of the nucleus and DCFH-DA. (**A**,**B**) Confocal microscopy was used to visualize the effects of EEBP from acorn and darae on LPS-induced ROS generation by DCFH-DA staining. (**C**) The amount of ROS production is expressed as mean fluorescence intensity (DCFH-DA). Data for *n* = 3 are shown as mean ± SD. ### *p* < 0.001, compared with the control group.*** *p* < 0.001, compared with the LPS group.

**Figure 3 antioxidants-13-00981-f003:**
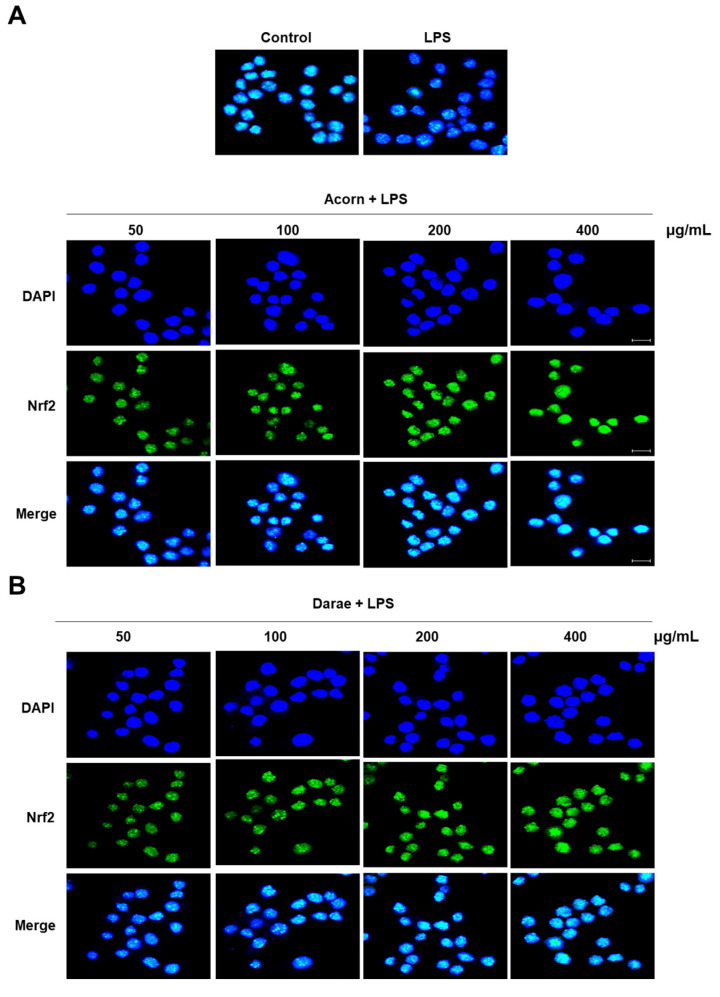
EEBP from acorn and darae induces nuclear translocation of Nrf2 in RAW 264.7 macrophages. Various amounts of bee pollens from acorn and darae were applied to the LPS-stimulated RAW264.7 cells (1 µg/mL). Cells were incubated for 24 h prior to the analysis. Using an immunofluorescence assay, the nuclear translocation of Nrf2 was investigated in cells treated with bee pollens from (**A**) acorn and (**B**) darae.

**Figure 4 antioxidants-13-00981-f004:**
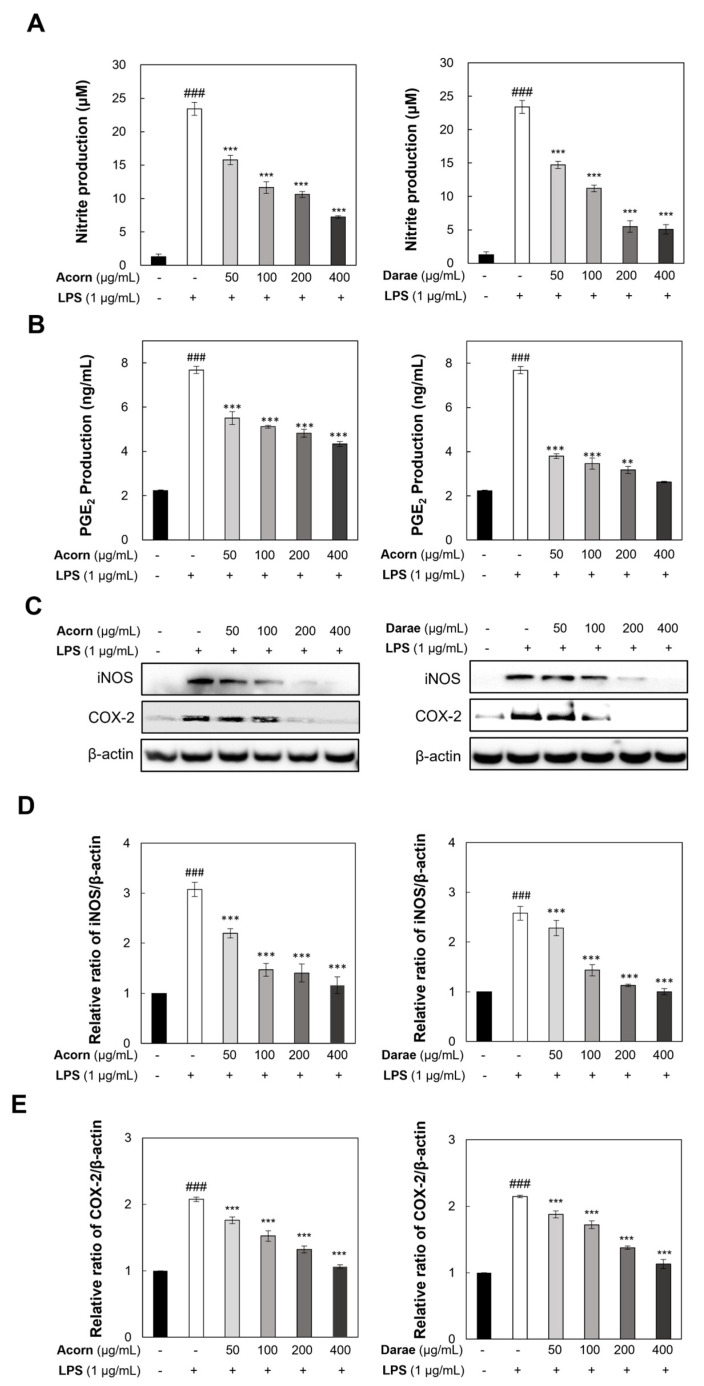
EEBP from acorn and darae inhibits pro-inflammatory mediators in LPS-stimulated RAW 264.7 macrophages. (**A**) The Griess reaction was used to quantify the nitrite content. (**B**) A commercial ELISA kit was used to assess the PGE2 levels in the culture medium. (**C**) Representative Western blot analysis was performed to measure iNOS and COX-2 expression. (**D**,**E**) The quantification of Western blot analysis by pollens from acorn and darae in LPS-stimulated RAW 264.7 macrophages. Every value represents the mean ± standard deviation and reflects the findings from three different experiments. ### *p* < 0.001, compared with the control group. ** *p* < 0.01, and *** *p* < 0.001, compared with the LPS group.

**Figure 5 antioxidants-13-00981-f005:**
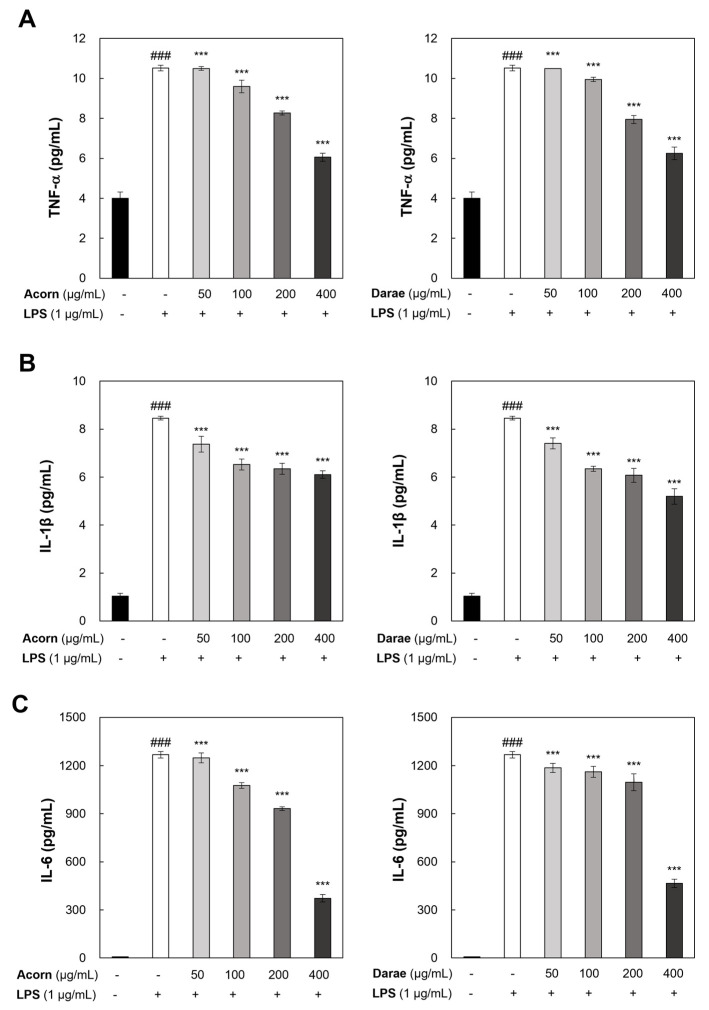
EEBP from acorn and darae reduces LPS-stimulated pro-inflammatory cytokines in RAW 264.7 macrophages. After the 24-h incubation period, (**A**) TNF-α, (**B**) IL-1β, and (**C**) IL-6 concentrations in cell-free supernatants were determined by ELISA. Every value represents the mean ± standard deviation and reflects the findings from three separate studies. ### *p* < 0.001, compared with the control group. *** *p* < 0.001, compared with the LPS group.

**Figure 6 antioxidants-13-00981-f006:**
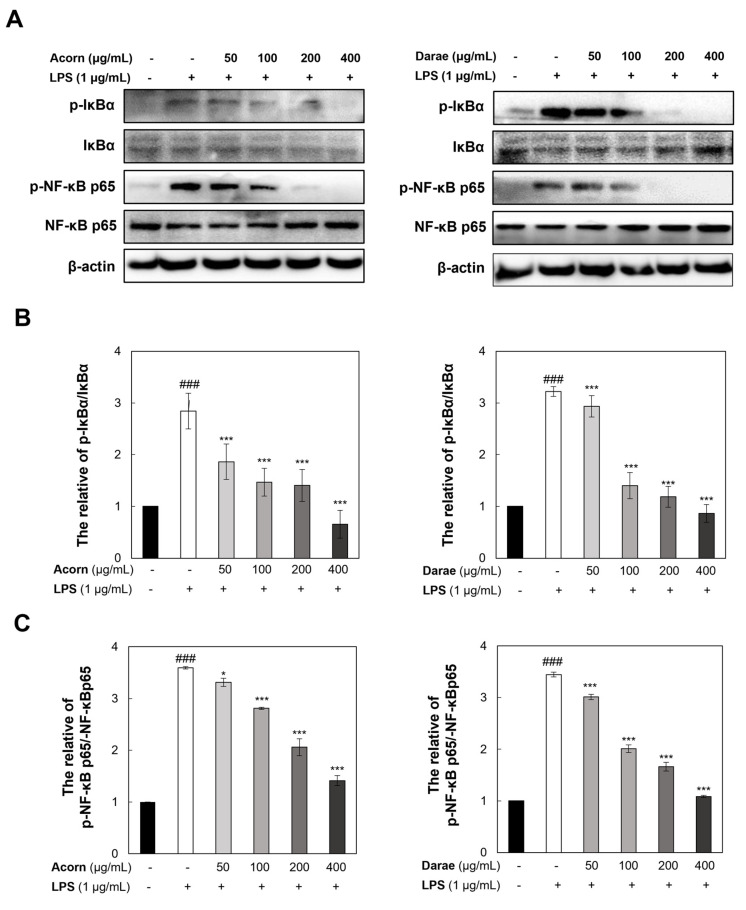
LPS-induced NF-κB p65 nuclear translocation is induced by EEBP from acorn and darae. Western blot analyses were used to examine the levels of p65 and IkB (**A**–**C**). Cells were treated with and without EEBP from acorn and darae, fixed, and subjected to total p65 antibody immunostaining (green) counterstained with DAPI for nuclei (blue). Confocal microscopy was used to obtain fluorescence images of EEBP treatment group from (**D**) acorn and (**E**) darae. Scale bar: 400 μm. The three experiments’ mean ± SD data are displayed. ### *p* < 0.001, compared with the control group. * *p* < 0.05 and *** *p* < 0.001, compared with the LPS group.

**Figure 7 antioxidants-13-00981-f007:**
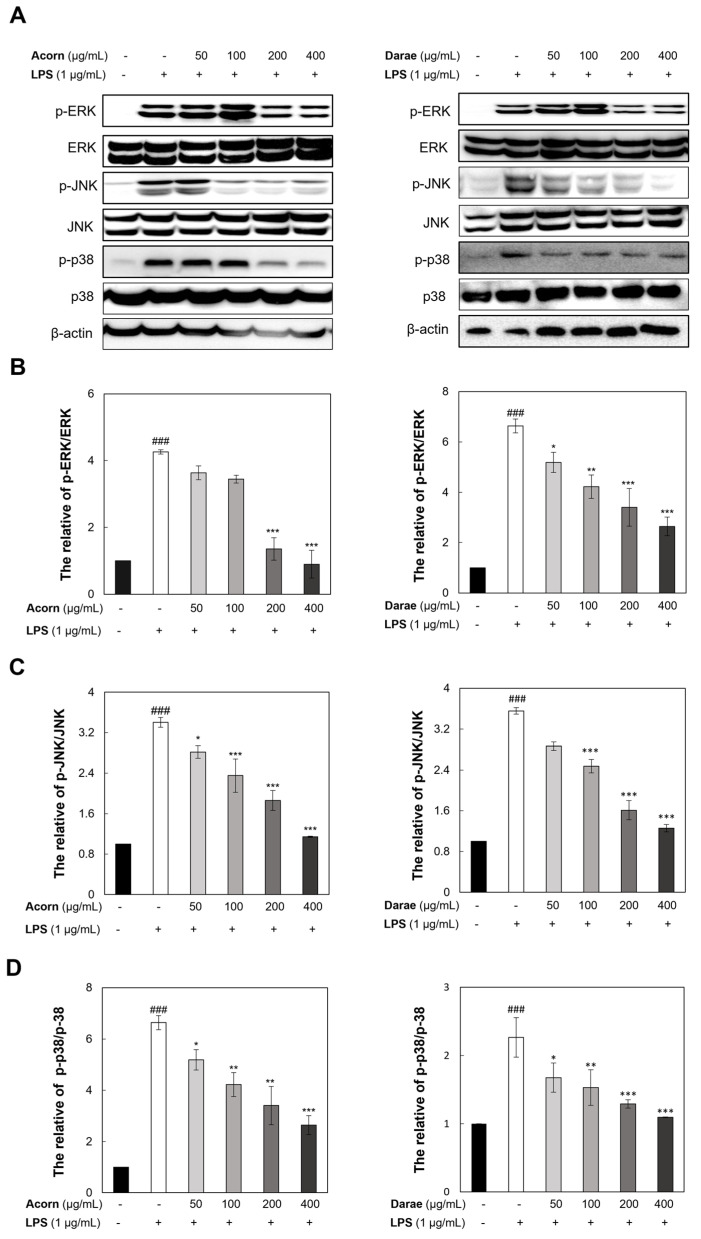
EEBP from acorn and darae diminishes LPS-stimulated activation of MAPKs in RAW 264.7 macrophages. RAW 264.7 macrophages were pretreated with EEBP from acorn and darae for 24 h, followed by the LPS treatment. The protein levels of ERK, JNK, and p38 were measured using Western blot experiments. Representative results are displayed in (**A**). Quantification of Western blot analysis of phosphorylation of (**B**) ERK, (**C**) JNK, and (**D**) p38 in cytoplasmic proteins is shown. The data are the densitometric value mean ± SD (*n* = 3) adjusted using β-actin. ### *p* < 0.001, compared with the control group. * *p* < 0.05, ** *p* < 0.01, and *** *p* < 0.001, compared with the LPS group.

**Figure 8 antioxidants-13-00981-f008:**
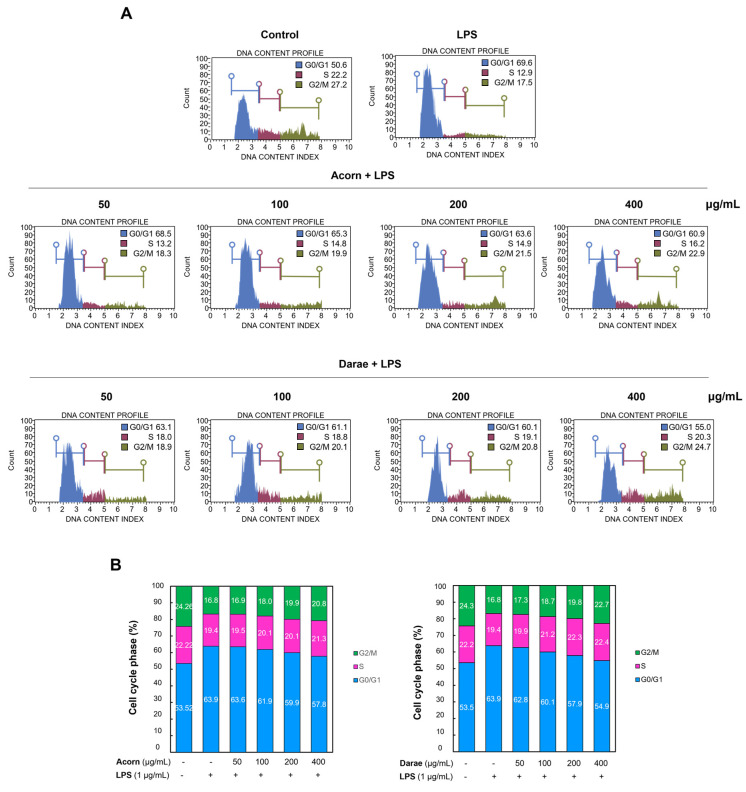
EEBP from acorn and darae induces cell cycle arrest in LPS-stimulated RAW 264.7 macrophages. (**A**) The cell cycle of LPS-stimulated RAW 264.7 cells with or without EEBP from acorn and darae was assessed using flow cytometry. (**B**) The percentage of apoptotic cells relative to the total cell population is quantified. Data for *n* = 3 are shown as mean ± SD.

## Data Availability

All of the data is contained within the article.

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
