# Peer review of "The Antioxidant and Anti-Inflammatory Properties of Bee Pollen from Acorn (Quercus acutissima Carr.) and Darae (Actinidia arguta)"

_antioxidants, 2024, doi:10.3390/antiox13080981_

Round 1

Reviewer 1 Report

The manuscript entitled “The Antioxidant and Anti-inflammatory Properties of Bee Pollen from Acorn (Quercus Acutissima Carr.) and Darae (Actinidia arguta)” have demonstrated significant results including estimation of biological activity (prevention of free radical production and anti-inflammatory) of bee pollen. The manuscript introduces new knowledge about biological activities of bee pollen.

Authors should correct manuscript according to the suggestion.

General comments:

ü  The Authors included "antioxidants" in the title, but according to what is included in the manuscript, the research relates rather to the prevention of the formation of free radicals. The Authors do not investigate the antioxidant mechanism of bee polled. Therefore suggests changing the title e.g."Prevention of free radicals and Anti-inflammatory Properties of Bee Pollen from Acorn (Quercus Acutissima Carr.) and Darae (Actinidia arguta)".

ü  There are no controls described in the Materials and Methods section. Authors must write what was the control (positive and negative) in individual experiments

ü  the legend for Figures 2-7 does not specify ###, *, ** and ***, this should be completed

ü  Figure 1 is there twice, but Figure 3 is missing

ü  The Discussion needs to be improved. There is no discussion of the obtained results with the information available in the literature. For example, what mechanism did other compounds of natural origin have, e.g. other types of bee pooled or phytochemicals? Moreover, the Authors describe only NF-kβ regulations based on their results, it is necessary to complete the description and compare this mechanism also with other compounds of natural origin

Introduction:

Line 66-67: please explain the abbreviations COX-2 and TNF-α

Materials and methods:

Line 104: give a % concentration for ethanol

Line 117-127: Please describe the mechanism of this assay, it was a determination of the degree of cell proliferation?, an assessment of mitochondrial metabolism?, or an assessment of the integrity of the cell membrane?

Line 132: why ROS production was estimated after 72 h incubation? was screening performed before? why was ROS production estimated after 72 h incubation? was screening performed earlier?.

Line 132 and 185: Please complete the concentrations of bee pollen

Line 216: It was biological or technical repetition? How many independent experiments carried out and witch repetition?

Author Response

Manuscript details:

Journal: Antioxidants

Manuscript ID: antioxidants-3107651

Title: The Antioxidant and Anti-inflammatory Properties of Bee Pollen from Acorn (Quercus Acutissima Carr.) and Darae (Actinidia argute)

Date: 23 July 2024

Answer to reviewers:

We are thankful for the reviewers’ constructive comments that helped to considerably improve and clarify the manuscript. We hope that its revised version answers their concerns. In the following we illustrate how we took the reviewers’ comments into account. Each reviewer is addressed individually, with the reviewer’s comments in italic font, our answers in normal font. We also made changes to the manuscript that is independent of the reviewers’ comments. Major changes we have made are as follows:

  1. The Authors included "antioxidants" in the title, but according to what is included in the manuscript, the research relates rather to the prevention of the formation of free radicals. The Authors do not investigate the antioxidant mechanism of bee polled. Therefore suggests changing the title e.g."Prevention of free radicals and Anti-inflammatory Properties of Bee Pollen from Acorn (Quercus Acutissima Carr.) and Darae (Actinidia arguta)".

Response:  We thank you for this appropriate comment. In this study, we observed that intercellular reactive oxygen species (ROS) formation by detecting by 2′,7′-Dichlor ofluorescindiacetate (DCF-DA), which is oxidized to fluorescent 2′,7′- Dichlorofluorescin (DCF) by oxidative stress. Also, For evaluation of NO production in the culture supernatant, nitrite, an oxidative product of NO, was measured by Griess reagent. Through this experiment, the results prove the prevention of free radicals such as ROS by ethanol extract bee pollens (EEBP) from acorn and darae. Thus, we agree/disagree with your suggestion of changing the title. to "Prevention of free Radicals and Anti-inflammatory Properties of Bee Pollen from Acorn (Quercus acutissima Carr.) and Darae (Actinidia arguta)".

  1. There are no controls described in the Materials and Methods section. Authors must write what was the control (positive and negative) in individual experiments.

Response:  We thank the reviewer for for kind suggestion. To reply to the comment, we revised the manuscript below:  

[Page 4, Line 178]

The positive control was cells that were not treated, and the negative control was LPS.

[Page 4, Line 189]

The positive control was cells that were not treated, and the negative control was LPS. We have checked the ROS production time-dependently (24, 48, 72 h).

[Page 5, Line 200; Page 5, Line 216; Page 5, Line 226; Page 5, Line 236; Page 6, Line 247; Page6, Line 255; Page 6, Line 270]

The positive control was cells that were not treated, and the negative control was LPS.

  1. the legend for Figures 2-7 does not specify ###, *, ** and ***, this should be completed.

Response:  Thank you for your comment. Following the comment, we revised the manuscript as below:

[Page 11, Line 360; Page 13, Line 376; Page 15, Line 391; Page 17, Line 414; Page 19, Line 429]

### p < 0.001, compared with the control group. *p < 0.05, **p < 0.01, ***p < 0.001, compared with the LPS group.

  1. Figure 1 is there twice, but Figure 3 is missing.

Response:  Thank you for your comment. We double-check whether Figure 3 is located in the manuscript.

  1. The Discussion needs to be improved. There is no discussion of the obtained results with the information available in the literature. For example, what mechanism did other compounds of natural origin have, e.g. other types of bee pooled or phytochemicals? Moreover, the Authors describe only NF-kβ regulations based on their results, it is necessary to complete the description and compare this mechanism also with other compounds of natural origin.

Response: We thank you for your astute observations regarding the discussion section. We revised the manuscript following your comment:

[Page 21, Line 519-525]

 In a recent paper, when treated with ethanol-extracted acorns (EeA), they inhibited the expression of interleukin (IL)-8 stimulated by induction nitric oxide synthase (iN-OS), cyclooxygenase 2 (COX2), monocyte chemoattractant protein-1 (MCP-1), and tumor necrosis factor (TNF) -α in human adult low calcium (HaCaT) cells [4715]. The difference from our paper is the additional investigation of the monocyte chemo-trained protein-1 (MCP-1) pathway, a type of chemokine that moves and attaches monocytes, which play an important role in the early stages of atherosclerosis as well as anti-inflammation.

  1. Line 66-67: please explain the abbreviations COX-2 and TNF-α.

Response:  We are grateful for your meticulous attention. We revised the manuscript following your comment:

[Page 3, Line 103,104]

COX-2: Cyclooxygenase-2

TNFα: Tumor Necrosis Factor-alpha

  1. Line 104: give a % concentration for ethanol.

Response:  We are grateful for your meticulous attention. We revised the manuscript following your comment:

[Page 4, Line 158] 70% ethanol

  1. Line 117-127: Please describe the mechanism of this assay, it was a determination of the degree of cell proliferation?, an assessment of mitochondrial metabolism?, or an assessment of the integrity of the cell membrane?

Response: We extend our gratitude to you for your detailed review. We revised the manuscript following your comment:

[Page 4, Line 174].

Cell Counting Kit-8 (CCK-8) allows sensitive colorimetric assays for the determination of cell viability in cytotoxicity assays.

  1. Line 132: why ROS production was estimated after 72 h incubation? was screening performed before? why was ROS production estimated after 72 h incubation? was screening performed earlier?

Response: We extend our gratitude to you for your detailed review. We have made changes to the manuscript in line with your suggestions:

[Page 5, Line 192].

We have checked the ROS production time-dependently (24, 48, 72 h). Based on that, we chose one time point, 72h.

  1. Line 132 and 185: Please complete the concentrations of bee pollen.

Response: We deeply value your detailed assessment. In response to your suggestions, we have revised the manuscript:

[Page 4, Line 190]

RAW 264.7 cells were seeded at 1 × 104 cells/well in 24-well, incubated for 24 h, and further incubated during 72 h with various concentrations (50, 100, 200, 400 μg/mL) of pollen samples

 [Page 6, Line 250]

Briefly, RAW 264.7 cells were plated into a 48-well plate (1×106 cells/well) overnight; the cells were pre-treated with pollen samples (50, 100, 200, 400 μg/mL) for 1h and then treated with 1 µg/mL of LPS for 18 h.

  1. Line 216: It was biological or technical repetition? How many independent experiments carried out and witch repetition?

Response: We extend our gratitude to you for your detailed review. We incorporated your recommendations into the manuscript:

[Page 6, Line 285]

Throughout this study, we performed at least three independent experiments (for biological repetition) with duplicated samples (for technical repetition).

We appreciate all of the helpful suggestions from the reviewers. They help us improve the quality of this paper. We hope you find that this manuscript meets the criteria for publication in the Nutrients.

Sincerely yours,

Mok-Ryeon Ahn

Department of Food Science & Nutrition

Dong-A University

Busan 49315, Korea

Tel.: +82-51-200-7324

Fax: +82-51-200-7535

Reviewer 2 Report

Manuscript details:
Journal: Antioxidants
Manuscript ID: antioxidants-3107651
Type of manuscript: Article
Title: The Antioxidant and Anti-inflammatory Properties of Bee Pollen from Acorn (Quercus Acutissima Carr.) and Darae (Actinidia arguta)

July 3, 2024

This is an interesting study that delves into investigating the antioxidant and anti-inflammatory properties of been pollen from acorn and darae pollens found in Korea. I think overall, the manuscript is well written with some suggestions for improvements:

1. Introduction: Define aging with the context of inflammatory disease to allow transition between the first few sentences in the first paragraph.

2. The introduction makes a compelling case for this study as it was mentioned that NSAIDs are known to have disadvantages including stomach ulcers, elevated blood sugar, and indications of steroid withdrawal. Please try to clarify what is meant by steroid withdrawal. Further, there should be more evidence of other previous related studies and emphasize on the novelty of the study. Upon cursory search in scholar.google.com, the following related studies are available but I think none were cited in the manuscript:

·       Algethami, J.S.; El-Wahed, A.A.A.; Elashal, M.H.; Ahmed, H.R.; Elshafiey, E.H.; Omar, E.M.; Naggar, Y.A.; Algethami, A.F.; Shou, Q.; Alsharif, S.M.; et al. Bee Pollen: Clinical Trials and Patent Applications. Nutrients 202214, 2858. https://doi.org/10.3390/nu14142858

·       Ali, A.M.; Kunugi, H. Apitherapy for Age-Related Skeletal Muscle Dysfunction (Sarcopenia): A Review on the Effects of Royal Jelly, Propolis, and Bee Pollen. Foods 20209, 1362. https://doi.org/10.3390/foods9101362

·       Gercek, Y.C.; Celik, S.; Bayram, S. Screening of Plant Pollen Sources, Polyphenolic Compounds, Fatty Acids and Antioxidant/Antimicrobial Activity from Bee Pollen. Molecules 202227, 117. https://doi.org/10.3390/molecules27010117

·       Sun L, Guo Y, Zhang Y and Zhuang Y (2017) Antioxidant and Anti-tyrosinase Activities of Phenolic Extracts from Rape Bee Pollen and Inhibitory Melanogenesis by cAMP/MITF/TYR Pathway in B16 Mouse Melanoma Cells. Front. Pharmacol. 8:104. doi: 10.3389/fphar.2017.00104

·       Sahin and Karkar (2019) The antioxidant properties of the chestnut bee pollen extract and its preventive action against oxidatively induced damage in DNA bases. Journal of Food Biochemistry. https://doi.org/10.1111/jfbc.12888

·       Ali, A.M.; Kunugi, H. Royal Jelly as an Intelligent Anti-Aging Agent—A Focus on Cognitive Aging and Alzheimer’s Disease: A Review. Antioxidants 20209, 937. https://doi.org/10.3390/antiox9100937

Please consider an extensive citation of previous available studies and how this research is novel, distinct, and has evolved from previously published ones. In this regard, I would probably consider elaborating more the introduction as it appears to be short.

3. Elaborate the significance of this study given that the aforementioned pollens are known to be abundant in Korea. Would there be implications from the economic standpoint? Would there be any related studies published in these kinds of pollen?

4. The materials and methods section are very detailed, and I appreciate reading this section.

5. The English language for this manuscript is fine. The statistical tests done are fine as well.

6. In figure 1 and other figures, please emphasize or highlight the important aspects of the figures.

7. Overall, the manuscript is interesting and I recommend this for publication. I think it will be beneficial to have a figure of the proposed mechanism for the anti-oxidant and anti-inflammatory properties of the pollens.

8. In the introduction, try to compare and contrast the two pollens.

9. Please consider adding more recent manuscripts in the references. Most of these references are outdated.

I compiled the detailed comments in the major comments section.

Author Response

Manuscript details:

Journal: Antioxidants

Manuscript ID: antioxidants-3107651

Title: The Antioxidant and Anti-inflammatory Properties of Bee Pollen from Acorn (Quercus Acutissima Carr.) and Darae (Actinidia argute)

Date: 23 July 2024

Answer to reviewers:

We are thankful for the reviewers’ constructive comments that helped to considerably improve and clarify the manuscript. We hope that its revised version answers their concerns. In the following we illustrate how we took the reviewers’ comments into account. Each reviewer is addressed individually, with the reviewer’s comments in italic font, our answers in normal font. We also made changes to the manuscript that is independent of the reviewers’ comments. Major changes we have made are as follows:

  1. Introduction: Define aging with the context of inflammatory disease to allow transition between the first few sentences in the first paragraph.

Response:  We thank you for this appropriate comment. To reply to the comment, we revised the text in the revised manuscript as follows:

[Page 1, Lines 37-45] Aging is a multifaceted biological process that entails a gradual reduction in physical function. With increasing life expectancy and the advancement and enrichment of society, the prevalence of these inflammatory diseases is expected to rise.  This process also leads to an increased vulnerability to age-related chronic diseases such as cardiovascular diseases, cancer, and neurodegenerative disorders [1]. The phenomenon of age-associated chronic inflammation, termed ‘senoinflammation,’ has garnered significant attention [2]. These chronic inflammations are closely linked to aging and age-related diseases. Aging induces changes at molecular, cellular, and systemic levels in inflammation, resulting in senoinflammation [2].

  1. The introduction makes a compelling case for this study as it was mentioned that NSAIDs are known to have disadvantages including stomach ulcers, elevated blood sugar, and indications of steroid withdrawal. Please try to clarify what is meant by steroid withdrawal. Further, there should be more evidence of other previous related studies and emphasize on the novelty of the study. Upon cursory search in scholar.google.com, the following related studies are available but I think none were cited in the manuscript:

  • Algethami, J.S.; El-Wahed, A.A.A.; Elashal, M.H.; Ahmed, H.R.; Elshafiey, E.H.; Omar, E.M.; Naggar, Y.A.; Algethami, A.F.; Shou, Q.; Alsharif, S.M.; et al. Bee Pollen: Clinical Trials and Patent Applications. Nutrients 2022, 14, 2858. https://doi.org/10.3390/nu14142858
  • Ali, A.M.; Kunugi, H. Apitherapy for Age-Related Skeletal Muscle Dysfunction (Sarcopenia): A Review on the Effects of Royal Jelly, Propolis, and Bee Pollen. Foods 2020, 9, 1362. https://doi.org/10.3390/foods9101362
  • Gercek, Y.C.; Celik, S.; Bayram, S. Screening of Plant Pollen Sources, Polyphenolic Compounds, Fatty Acids and Antioxidant/Antimicrobial Activity from Bee Pollen. Molecules 2022, 27, 117. https://doi.org/10.3390/molecules27010117
  • Sun L, Guo Y, Zhang Y and Zhuang Y (2017) Antioxidant and Anti-tyrosinase Activities of Phenolic Extracts from Rape Bee Pollen and Inhibitory Melanogenesis by cAMP/MITF/TYR Pathway in B16 Mouse Melanoma Cells. Front. Pharmacol. 8:104. doi: 10.3389/fphar.2017.00104
  • Sahin and Karkar (2019) The antioxidant properties of the chestnut bee pollen extract and its preventive action against oxidatively induced damage in DNA bases. Journal of Food Biochemistry. https://doi.org/10.1111/jfbc.12888 2
  • Ali, A.M.; Kunugi, H. Royal Jelly as an Intelligent Anti-Aging Agent—A Focus on Cognitive Aging and Alzheimer’s Disease: A Review. Antioxidants 2020, 9, 937. https://doi.org/10.3390/antiox9100937

Please consider an extensive citation of previous available studies and how this research is novel, distinct, and has evolved from previously published ones. In this regard, I would probably consider elaborating more the introduction as it appears to be short.

Response:  We sincerely appreciate your thorough review. Based on your suggestion, we try to emphasize the significance of this study. To reply to the comment, we newly added the text in the revised manuscript as follows:

[Page 2, Lines 46-68] Substances derived from natural sources, such as bee pollen extract and phyto-chemicals, show great promise as anti-inflammatory treatments with reduced side ef-fects of current therapeutics [3]. Currently, corticosteroids and non-steroidal an-ti-inflammatory medications (NSAIDs) are frequently used in conventional therapy regimens [4]. Nevertheless, these options have disadvantages, such as organ damage [4]. When long-term use of corticosteroid medications, such as prednisone, is suddenly discontinued, the body may not be able to produce hormones adequately, leading to steroid withdrawal symptoms like stomach ulcers and elevated blood sugar. This drives the quest for natural substances that have fewer adverse effects while simulat-ing the effects of NSAIDs [5].

Bee pollen could be a promising alternative to current therapeutics like NSAIDs or corticosteroids by addressing current medical unmet needs. Recent studies confirm the potential of bee pollen or other bee products as antioxidants and an-ti-inflammatory medicines [6, 7]. Bee pollen is a mixture of nectar and secretions from honeybees combined with floral pollen. It can be collected at the hive entrance using traps [8]. Due to its anti-disease properties, bee pollen is incorporated into diets as an additional source of nutrients. It contains essential nutrients, including carbohydrates, proteins, fats, vitamins, and minerals, as well as trace elements and high concentra-tions of polyphenols, predominantly flavonoids [9]. The current research reported the antioxidant and anti-inflammatory effects of bee pollens [3, 6, 7], implying the potential of bee pollen for the development of anti-oxidant and anti-inflammation therapy without significant side effects [6, 7]. It has played an important role in combating metabolic disorders such as diabetes, obesity, hyperlipidemia, and related cardiovascular complications as well as anti-inflammatory antioxidants [10].

[Page 2-3, Lines 95-108]

Numerous studies have investigated bee pollen, yet the variety of the origin plant types and their regional specificities remain still underexplored. It is the first scientific report of these effects of bee pollens, specifically from acorn and darae. To our knowledge, no other research has focused specifically on bee pollen from acorn and darae. Previous reports have demonstrated the anti-oxidant and anti-inflammatory effects of bee pollen worldwide [19‒22]. However, these results indicate the ef-fects of bee pollen vary slightly. This variation may be attributed to the nutritional and chemical properties of bee pollen, which are influenced by the origin plants and har-vest regions. Previously, we analyzed the chemical properties of Korean bee pollen and found that they differ depending on the origin plants and harvest regions [23]. Based on our ongoing nutritional analysis (ongoing preparation for publication), components and nutritional content of bee pollen vary with plant species and harvest regions. We speculate that nutritional and chemical components and nutrition of bee pollen ac-count for their various effects and characteristics of bee pollen.

  1. Elaborate the significance of this study given that the aforementioned pollens are known to be abundant in Korea. Would there be implications from the economic standpoint? Would there be any related studies published in these kinds of pollen?

Response:  We appreciate your thorough and insightful feedback. Regarding economic feasibility, there was information on the economic value of pollination, but nothing specifically about bee pollen itself. The review paper only contains information on the benefits, effects, and potential uses of bee pollen. The manuscript has been modified according to your comments: [Page 21, Lines 506-517]

Despite the extensive research on bee pollen, the diverse origin plant types of pollen and regional characteristics have not been thoroughly examined. This study investigated the characteristics of bee pollen from acorn and darae in South Korea in anti-oxidation and anti-inflammation. We previously researched the chemical properties of Korean bee pollen and their catechol‑O‑methyltransferase inhibitory activities, indicating their potential prevention and treatment of Parkinson’s disease and depression [42]. This study broadens our understanding by demonstrating the effects of bee pollen from acorn and darae, supporting their development as an effective anti-inflammatory and anti-oxidant treatment. Enhancing the potential development of bee pollen can stimulate their consumption and use in functional foods and medicine. Consequently, this research will benefit the apiculture industry and the local economy. Therefore, this study will contribute to both scientific knowledge and the local and apiculture industry.

  1. The materials and methods section are very detailed, and I appreciate reading this section.

Response:  We appreciate your insightful feedback and favorable assessment of our manuscript. We are glad to know that you are satisfied with the materials and methods section.

  1. The English language for this manuscript is fine. The statistical tests done are fine as well.

Response:  Thank you for your thoughtful comment and positive evaluation of our manuscript. We are pleased to hear that you find the language and statistical analyses to be satisfactory.

  1. In figure 1 and other figures, please emphasize or highlight the important aspects of the figures.

Response:  We thank you for your astute observations. We value the reviewer’s contribution to enhancing the quality of our paper. The manuscript has been modified according to your comments:

Figure 1

[Page 6, Lines 286-287] Figure 1 showed that EEBP from both acorn and darae has no negative effects on the cell viability of RAW 264.7 cells.

Figure 2.

[Page 8, Lines 309-311] These data demonstrated that LPS-induced ROS generation was inhibited by acorn and darae, implying the therapeutic effects of EEBP from acorn and darae as an anti-oxidant (Figure 2A,B).

Figure 3.

[Page 8, Lines 320-323] Specifically, after the administration of EEBP from acorn and darae, Nrf2 accumulation was observed in the nucleus, whereas in LPS-treated cells, Nrf2 expression was predominantly in the cytoplasm (Figure 3A & B). These findings strongly suggest that EEBP from acorn and darae promotes Nrf2 activation and nuclear translocation in macrophages

Figure 4.

[Page 8, Lines 340-342] According to these findings in Figure 4, EEBP from acorn and darae was able to suppress the expression of the iNOS and COX-2 enzymes, resulting in consequently decreased generation of NO and PGE2.

Figure 5.

[Page 13, Lines 373-376] As a result, data shown in Figure 5 illustrates that the pre-treatment with EEBP from acorn and darae significantly decreased the release of TNF-α, IL-1β, and IL-6 in a concentration-dependent manner, whereas the challenge with LPS alone caused increases in the levels of these three cytokines.

Figure 6.

[Page 8, Lines 389-396] Figure 6A–C demonstrates that LPS-stimulated cells showed higher expression levels of the proteins p65 and IκB-α. However, LPS-stimulated nuclear accumulation of p65 and IκB-α was significantly diminished by treatment with EEBP from acorn and darae in a concentration-dependent manner.

Immunocytochemistry analysis also revealed that NF-κB p65 was typically sequestered in the cytoplasm under LPS treatment, which is consistent with Western Blot analysis (Figure 6A–C). However, pretreatment with EEBP from acorn and darae significantly inhibited LPS-mediated nuclear translocation of NF-κB (Figure 6D,E).

Figure 7.

[Page 8, Lines 409-411] Phosphorylation of three of the MAPK proteins, ERK, JNK, and p38, were upregulated by LPS stimulation (Figure 7). Furthermore, EEBP from acorn and darae reduced the LPS-induced phosphorylation of ERK, JNK, and p38 MAPK in a concentration-dependent manner.

Figure 8.

[Page 18, Lines 422-428] Figure 8 demonstrated that Tthe S phase decreased from 22.2% to 19.4%, the G1 phase increased from 53.5% to 63.9%, and the G2/M phase reduced from 24.26% to 16.8% in both groups' LPS treatment groups with LPS-induced inflammation. These findings suggested that G1 arrest could result from an inflammatory response triggered by LPS. However, cells pre-treated with bee pollens from acorn and darae exhibited a dose-dependent decrease in the proportion of G1 cycles.

  1. Overall, the manuscript is interesting and I recommend this for publication. I think it will be beneficial to have a figure of the proposed mechanism for the anti-oxidant and anti-inflammatory properties of the pollens.

Response:  We appreciate your insightful feedback and your positive assessment of our manuscript. Here is the signaling pathway of how bee pollens from acorn and darae work in LPS-induced RAW267.4 cells.

  1. In the introduction, try to compare and contrast the two pollens.

Response:  We appreciate your thorough and insightful feedback. We have studied the bee pollen from acorn and darae, but we could not observe the significant differences between the two pollens. To reply to the comment, we revised the text in the revised manuscript

[Page 95, Line 472]

However, there is no significant difference in the anti-oxidant and anti-inflammatory effects of EEBP from acorn and darae.

  1. Please consider adding more recent manuscripts in the references. Most of these references are outdated.

Response:  Thank you for suggesting recent references. We have replaced older papers with newer ones, updating our selection to include 26 papers from 2019 onwards, which now constitute more than half of the total references. Please refer to the revised manuscript for details [Page 20-22, Lines 544-649].

We appreciate all of the helpful suggestions from the reviewers. They help us improve the quality of this paper. We hope you find that this manuscript meets the criteria for publication in the Nutrients.

Sincerely yours,

Mok-Ryeon Ahn

Department of Food Science & Nutrition

Dong-A University

Busan 49315, Korea

Reviewer 3 Report

The work seems to framed into a too old scientific environments and it is not properly justified in light of recent developments on this field.

Title, Abstract, and Keywords:

The title relates well with the described work, and is brief but elucidative.

The abstract is well organized but lacks some quantitative results. Just an  example when saying “it reduced active oxygen species (ROS) production and  increased the nuclear potential of …” it is important to have an idea of how much was it a reduction of 1-2%, or 20% or 60%. It is relevant to quantity some of the results to give credit to the conclusions.

Introduction:

The introduction helps to frame and contextualize the work. It presents some state of the art on a number of topics related to the research topic, but very widespread, not focused on the resrecah property. In fact, it does not make a proper state of the art of the most important topic of the research, and it only presents one sentence: “The current research reported the antioxidant and anti-inflammatory effects of bee pollens  [3], implying the potential of bee pollen for the development of anti-oxidant and anti-inflammation therapy.” How many studies have already been developed on theses topics?

For example on pubmed I found several studies focusing the bee pollen antioxidant and anti-inflammatory effect, but I just cite a few: DOI: 10.4103/1735-5362.278716, DOI: 10.1007/s10787-023-01182-4, DOI: 10.3390/ijms20184512, DOI: 10.3390/ijms20184512, DOI: 10.1080/14786419.2023.2172727, etc…..

AS so, it is not clear why this reserach is justified.

Materials and methods:

The description of the methodologies applied to conduct experiments and laboratorial essays is clearly presented, with detail enabling possible replication.

I have no recommendations on this part.

Results:

The presentation of results is clear, in the form of Figures and Tables, appropriate for each case. The results are easily understandable form the presented elements.

However, I find one exception, because the two graphs in Figure 8 are not clear in what concerns the labels, because the numbers are in black colour and the bars are all too dark. So I recommend to make the colour of the bars clearer (clear blue, clear red – pink, clear green) or in alternative maintain the colour of the bars and change the letters to white for better contrast, and easy reading.

Discussion:

The discussion could benefit from a more exhaustive comparison of the results with other published  works also for bee pollen, because the discussion is not strng from this point of view.

Conclusions

The conclusions part is also well enough, presenting the most relevant findings of the work. I do not have reccomendations

References

There is an unacceptable rate of recent references. In 37 references only 8 are from the last 5 years (2019 or after), representing only 22%. Additionally, there are several references from the last century from the nineties, which are 20 or 30 years old. This reveals a lack of knowledge about what has been done in the scientific community in the present, and it is inexcusable to not know what has recently been investigated. For that matter the researchers may possibly have investigated what was already investigated by other authors 2 or 3 years ago. This was already visible in the introduction and in the discussion, and it leaves a great doubt about the scientific validity of the work.

Author Response

Manuscript details:

Journal: Antioxidants

Manuscript ID: antioxidants-3107651

Title: The Antioxidant and Anti-inflammatory Properties of Bee Pollen from Acorn (Quercus Acutissima Carr.) and Darae (Actinidia argute)

Date: 23 July 2024

Answer to reviewers:

We are thankful for the reviewers’ constructive comments that helped to considerably improve and clarify the manuscript. We hope that its revised version answers their concerns. In the following we illustrate how we took the reviewers’ comments into account. Each reviewer is addressed individually, with the reviewer’s comments in italic font, our answers in normal font. We also made changes to the manuscript that is independent of the reviewers’ comments. Major changes we have made are as follows:

[Major comments]

  1. The work seems to framed into a too old scientific environments and it is not properly justified in light of recent developments on this field.

Response:  Thank you for your concerns regarding this study. While it may not be state-of-the-art research technique such as big data or AI, it is nonetheless significant in characterizing the anti-oxidant and anti-inflammatory effects of bee pollens from acorn and darae regionally produced in South Korea. It is the first scientific report of these effects of bee pollens, specifically from acorn and darae. To our knowledge, no other research has focused specifically on bee pollen from acorn and darae.

Numerous reports have demonstrated the anti-oxidant and anti-inflammatory effects of bee pollen worldwide [Eteraf-Oskouei et al., 2022, J Agric Food Chem.; Kosedag et al., 2023,. Inflammopharmacology.; Lopes et al., 2019, Int J Mol Sci.; Capparelli et al., 2024, Nat Prod Res]. However, these results indicate the effects of bee pollen vary slightly [Eteraf-Oskouei et al., 2022, J Agric Food Chem.; Kosedag et al., 2023, Inflammopharmacology.; Lopes et al., 2019, Int J Mol Sci.; Capparelli et al., 2024, Nat Prod Res]. It may be due to the nutritional and chemical properties of bee pollen, which are influenced by the origin plants and harvest regions. Previously, we analyzed the chemical properties of Korean bee pollen and found that they differ depending on the origin plants and harvest regions (Miyata R et al., 2022, J Agric Food Chem). Based on our ongoing nutritional analysis (ongoing preparation for publication), components and nutritional content of bee pollen vary with plant species and harvest regions. We speculate that nutritional and chemical components and nutrition of bee pollen account for their various effects and characteristics of bee pollen.

Additionally, expanding the research area on regional bee pollen is expected to benefit the bee-keeping industry and the local economy. We have analyzed the chemical properties of Korean bee pollen and their catechol‑O‑methyltransferase (COMT) inhibitory activities, indicating their potential prevention and treatment of Parkinson’s disease and depression (Miyata R et al., 2022, J Agric Food Chem). This study broadens the research field to investigate their other effects, such as anti-oxidation and anti-inflammation of bee pollen from acorn and darae. Enhancing the development potential of bee products can stimulate their use in the functional food and medicine. Consequently, this research will contribute to the bee-keeping industry and the local economy. Therefore, this traditional scientific study will shed light on the scientific knowledge and local economy.

  • Miyata, R; Hoshino, S.; Ahn, M-R.; Kumazawa, S. Chemical Profiles of Korean Bee Pollens and Their Catechol- O-methyltransferase Inhibitory Activities. J Agric Food Chem. 2022, 70(4):1174-1181.
  • Tahereh Eteraf-Oskouei, Ayda Shafiee-Khamneh, Fariba Heshmati-Afshar 2, Abbas Delazar 2. Anti-inflammatory and an-ti-angiogenesis effect of bee pollen methanolic extract using air pouch model of inflammation. Res Pharm Sci. 2020, 15(1):66-75.
  • Murat Kosedag, Mine Gulaboglu. Pollen and bee bread expressed highest anti-inflammatory activities among bee products in chronic inflammation: an experimental study with cotton pellet granuloma in rats. Inflammopharmacology. 2023. 31(4):1967-1975.
  • Alberto Jorge Oliveira Lopes, Cleydlenne Costa Vasconcelos, Francisco Assis Nascimento Pereira, Rosa Helena Moraes Silva, Pedro Felipe Dos Santos Queiroz, Caio Viana Fernandes, João Batista Santos Garcia 8, Ricardo Martins Ramos, Cláudia Quintino da Rocha, Silvia Tereza de Jesus Rodrigues Moreira Lima, Maria do Socorro de Sousa Cartágenes, Maria Nilce de Sousa Ribeiro. Anti-Inflammatory and Antinociceptive Activity of Pollen Extract Collected by Stingless Bee Melipona fasciculata. Int J Mol Sci. 2019. 20(18):4512.
  • Sonia Capparelli, Ylenia Pieracci, Francesca Coppola, Ilaria Marchioni, Simona Sagona, Antonio Felicioli, Luisa Pistelli, Laura Pistelli. The colors of Tuscan bee pollen: phytochemical profile and antioxidant activity. Nat Prod Res. 2024. 38(13):2313-2319

[Detail comments]

  1. Title, Abstract, and Keywords:

The title relates well with the described work, and is brief but elucidative.

The abstract is well organized but lacks some quantitative results. Just an example when saying “it reduced active oxygen species (ROS) production and increased the nuclear potential of …” it is important to have an idea of how much was it a reduction of 1-2%, or 20% or 60%. It is relevant to quantity some of the results to give credit to the conclusions.

Response:  Thank you for your advice regarding the abstract. We admit your suggestion and revise it. Please see the revised manuscript as follows:

[Page 1, Lines 12-32]

Abstract: Aging is a complex biological process characterized by a progressive decline in physical function and an increased risk of age-related chronic diseases. Additionally, oxidative stress is known to cause severe tissue damage and inflammation. Pollens from acorn and darae are extensively produced in Korea. However, the underlying molecular mechanisms of these components under the conditions of inflammation and oxidative stress remain largely unknown. This study aimed to investigate the effect of bee pollen components on lipopolysaccharide (LPS)-induced RAW 264.7 mouse macrophages. This study demonstrates that acorn and darae significantly inhibit the LPS-induced production of inflammatory mediators, such as nitric oxide (NO) and prostaglandin E2 (PGE2), in RAW 264.7 cells. Specifically, bee pollen from acorn reduces NO production by 69.23 ±0.04% and PGE2 production by 44.16% ± 0.08, while bee pollen from darae decreases NO pro-duction by 78.21 ± 0.06% and PGE2 production by 66.23 ± 0.1%. Furthermore, bee pollen form acorn and darae reduced active oxygen species (ROS) production by 47.01 ± 0.5% and 60 ± 0.9%, respectively. It increased the nuclear potential of nuclear factor erythroid 2-related factor 2 (Nrf2) in LPS-stimulated RAW 264.7 cells. Moreover, treatment with acorn and darae abolished the nuclear potential of nuclear factor κB (NF-κB) and reduced the expression of extracellular signal-associated kinase (ERK) and c-Jun N-terminal kinase (JNK) phosphorylation in LPS-stimulated RAW 264.7 cells. Specifically, acorn decreased NF-κB nuclear potential by 90.01 ± 0.3%, ERK phosphorylation by 76.19 ± 1.1%, and JNK phosphorylation by 57.14 ± 1.2%. Similarly, darae reduced NF-κB nuclear potential by 92.21 ± 0.5 %, ERK phosphorylation by 61.11 ± 0.8%, and JNK phosphorylation by 59.72 ± 1.12%. These results suggest that acorn and darae could be potential antioxidants and an-ti-inflammatory agents.

  1. Introduction:

The introduction helps to frame and contextualize the work. It presents some state of the art on a number of topics related to the research topic, but very widespread, not focused on the resrecah property. In fact, it does not make a proper state of the art of the most important topic of the research, and it only presents one sentence: “The current research reported the antioxidant and anti-inflammatory effects of bee pollens [3], implying the potential of bee pollen for the development of anti-oxidant and anti-inflammation therapy.” How many studies have already been developed on theses topics?

For example on pubmed I found several studies focusing the bee pollen antioxidant and anti-inflammatory effect, but I just cite a few: DOI: 10.4103/1735-5362.278716, DOI: 10.1007/s10787-023-01182-4, DOI: 10.3390/ijms20184512, DOI: 10.3390/ijms20184512, DOI: 10.1080/14786419.2023.2172727, etc…..

  AS so, it is not clear why this reserach is justified.

Response:  We thank you for this appropriate comment. To reply to the comment, we revised the text in the revised manuscript as follows:

[Page 2, Lines 55-68]

Bee pollen could be a promising alternative to current therapeutics like NSAIDs or corticosteroids by addressing current medical unmet needs. Recent studies confirm the potential of bee pollen or other bee products as antioxidants and anti-inflammatory medicines [6, 7]. Bee pollen is a mixture of nectar and secretions from honeybees combined with floral pollen. It can be collected at the hive entrance using traps [8]. Due to its anti-disease properties, bee pollen is incorporated into diets as an additional source of nutrients. It contains essential nutrients, including carbohydrates, proteins, fats, vitamins, and minerals, as well as trace elements and high concentrations of polyphenols, predominantly flavonoids [9]. The current research reported the antioxidant and anti-inflammatory effects of bee pollens [3, 6, 7], implying the potential of bee pollen for the development of anti-oxidant and anti-inflammation therapy without significant side effects [6, 7]. It has played an important role in combating metabolic disorders such as diabetes, obesity, hyperlipidemia, and related cardiovascular complications as well as anti-inflammatory antioxidants [10].

  1. Materials and methods:

The description of the methodologies applied to conduct experiments and laboratorial essays is clearly presented, with detail enabling possible replication. I have no recommendations on this part.

Response:  We are grateful for your kind comments and favorable evaluation of our manuscript. It is reassuring to hear that you find the materials and methods section adequate.

  1. Results:

The presentation of results is clear, in the form of Figures and Tables, appropriate for each case. The results are easily understandable form the presented elements.

  However, I find one exception, because the two graphs in Figure 8 are not clear in what concerns the labels, because the numbers are in black colour and the bars are all too dark. So I recommend to make the colour of the bars clearer (clear blue, clear red – pink, clear green) or in alternative maintain the colour of the bars and change the letters to white for better contrast, and easy reading.

      Response:  Thank you for your advice about Figure 8. To reply to this comment, we changed the colors to clarify the labels in Figure 8. Please see Figure 8 [Page 19, Line 446].

  1. Discussion:

The discussion could benefit from a more exhaustive comparison of the results with other published works also for bee pollen, because the discussion is not strng from this point of view.

Response:  We thank you for this appropriate comment. To reply to the comment, we revised the text in the revised manuscript as follows:

[Page 19, Lines 490-516]

Our results showed that EEBP from acorn and darae promotes ROS release in RAW264.7 cells, which, in turn, activates the nuclear translocation of Nrf2, exerting an anti-inflammatory effect. The activation of MAPK is known to induce Nrf2-mediated HO-1 expression [46]. Moreover, ROS has been reported to mediate the activation of various kinases, including MAPK [1]. The treatment of bee pollen from acorn and darae significantly inhibited LPS-induced phosphorylation of ERK, JNK, and p38 proteins in RAW264.7 cells. These results indicate that the effect of bee pollen from acorn and darae is mediated through Nrf2 activation via the ROS-dependent MAPKs pathway. In a recent paper, when treated with ethanol-extracted acorns (EeA), they inhibited the expression of interleukin (IL)-8 stimulated by induction nitric oxide synthase (iNOS), cyclooxygenase 2 (COX2), monocyte chemoattractant protein-1 (MCP-1), and tumor necrosis factor (TNF) -α in human adult low calcium (HaCaT) cells [47]. The difference from our paper is the additional investigation of the monocyte chemotrained protein-1 (MCP-1) pathway, a type of chemokine that moves and attaches monocytes, which play an important role in the early stages of atherosclerosis as well as an-ti-inflammation.

Despite the extensive research on bee pollen, the diverse origin plant types of pol-len and regional characteristics have not been thoroughly examined. This study inves-tigated the characteristics of bee pollen from acorn and darae in South Korea in an-ti-oxidation and anti-inflammation. We previously researched the chemical properties of Korean bee pollen and their catechol‑O‑methyltransferase inhibitory activities, indicating their potential prevention and treatment of Parkinson’s disease and depression [42]. This study broadens our understanding by demonstrating the effects of bee pollen from acorn and darae, supporting their development as an effective anti-inflammatory and anti-oxidant treatment. Enhancing the potential development of bee pollen can stimulate their consumption and use in functional foods and medicine. Consequently, this research will benefit the apiculture industry and the local economy. Therefore, this study will contribute to both scientific knowledge and the local and apiculture industry.

  1. Conclusions:

The conclusions part is also well enough, presenting the most relevant findings of the work. I do not have reccomendations.

Response:  Thank you for your considerate remarks and positive review of our conclusion. We are pleased that you find the conclusion section to be satisfactory.

  1. References:

There is an unacceptable rate of recent references. In 37 references only 8 are from the last 5 years (2019 or after), representing only 22%. Additionally, there are several references from the last century from the nineties, which are 20 or 30 years old. This reveals a lack of knowledge about what has been done in the scientific community in the present, and it is inexcusable to not know what has recently been investigated. For that matter the researchers may possibly have investigated what was already investigated by other authors 2 or 3 years ago. This was already visible in the introduction and in the discussion, and it leaves a great doubt about the scientific validity of the work.

Response:  Thank you for a good suggestion for recent references. We revised the papers, replacing old ones with recent ones, and updated the selection to include 26 papers from 2019 onwards, which now make up more than half of the total. Please see the reference part in our revised manuscript [Page 20-22, Lines 544-649].

We appreciate all of the helpful suggestions from the reviewers. They help us improve the quality of this paper. We hope you find that this manuscript meets the criteria for publication in the Nutrients.

Sincerely yours,

Mok-Ryeon Ahn

Department of Food Science & Nutrition

Dong-A University

Busan 49315, Korea

Round 2

Reviewer 1 Report

After correction manuscript can be accepted.

After correction manuscript can be accepted.